# Learning to Aggregate: A Parameterized Aggregator to Debias Model Aggregation for Cross-device Federated Learning

## Abstract

Federated learning (FL) collaboratively trains deep models on decentralized clients with privacy constraint. The aggregation of client parameters within a communication round suffers from the "client drift" due to the heterogeneity of client data distributions, resulting in unstable and slow convergence. Recent works typically impose regularization based on the client parameters to reduce the local aggregation heterogeneity for optimization. However, we argue that they generally neglect the inter-communication heterogeneity of data distributions ("period drift"), leading to deviations of intra-communication optimization from the global objective. In this work, we aim to calibrate the local aggregation under "client drift" and simultaneously approach the global objective under "period drift". To achieve this goal, we propose a learning-based aggregation strategy, named FEDPA, that employs a **P**arameterized **A**ggregator rather than non-adaptive techniques (*e.g.*, federated average). We frame FEDPA within a meta-learning setting, where aggregator serves as the meta-learner and the meta-task is to aggregate the client parameters to generalize well on a proxy dataset. Intuitively, the meta-learner is task-specific and could thereby acquire the meta-knowledge, *i.e.*, calibrating the parameter aggregation from a global view and approaching the global optimum for generalization.

## 1 Introduction

Federated Learning (FL) McMahan et al. (2017) has been an emerging privacy-preserving machine learning paradigm to collaboratively train a shared model on a decentralized manner without sharing private data. In FL, clients independently train the shared model over their private data, and the server aggregates the uploaded model parameters periodically until convergence. In FL Kairouz et al. (2021), a key challenge hindering effective model aggregation lies in the heterogeneous data of clients Zhao et al. (2018), especially in cross-device (as opposed to cross-silo) FL with a large amount of clients (e.g. mobile devices). Wherein, vanilla FL algorithms, such as federated averaging (FEDAVG) McMahan et al. (2017), based on averaging the parameters of candidate clients, would suffer from bad convergence and performance degradation.

Existing works Hsu et al. (2019); Li et al. (2020); Karimireddy et al. (2021) depict the non-iid trap as weight divergence Zhao et al. (2018) or client drift Karimireddy et al. (2021). To cope with it, they typically impose regularization in local optimization at each communication round such that the intra-round heterogeneity can be reduced. However, we argue that existing methods generally neglect the heterogeneity among different communication rounds, and the round-specific regularization would inevitably fall into a local optimum. Specifically, in cross-device FL, the sampled clients to be aggregated might involve different data distributions among different communication rounds. As such, the optimization direction estimated in a single round might deviate from that estimated with all clients, eventually amplifying the the aggregation bias[1], and resulting in bad convergence even oscillation. For simplicity, we term this challenge as "period drift", and provide empirical evidence in real-wolrd datasets (*c.f.* Figure 1).

---

[1] In ecological studies, aggregation bias is the expected difference between effects for the group and effects for the individual.

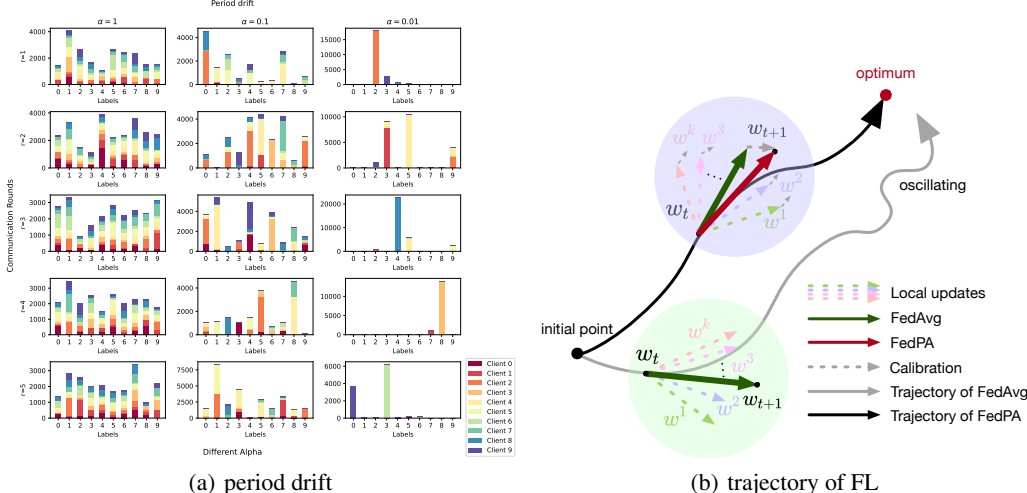

(a) period drift            (b) trajectory of FL

Figure 1: **Period drift in FL**. In the left figure, we give a example of a 10-class classification task with label distribution skew Kairouz et al. (2021). We consider three degrees of non-iidness by setting the dirichlet hyperparameter $\alpha = 1/0.1/0.01$ Hsu et al. (2019), and display the distribution difference of five communication rounds within a $5 \times 3$ grid. The colored blocks in the histogram represent the amount of data of different labels that belongs to the selected 10 clients of 100 clients. In a subfigure (i.e. within a communication round), client drift is exhibited by different colors, while in a column, the period drift can be presented by the length of bars. Period drift becomes more obvious as increasing the degree of non-iid (smaller $\alpha$). In the right figure, we illustrates the trajectory of FL, where the direction with period drift may deviates from the global optimum, resulting in slow convergence and oscillation, and how FEDPA calibrates and controls the trajectory of FL.

In this work, we set the goal of debiasing model aggregation under client drift and period drift in a unified framework. The key to this problem is to have an adaptive approximation of parameter calibration towards the global objective $f$ McMahan et al. (2017), which is, however, non-trivial to go beyond the local view and approach the optimum based on solely the intra-communication client parameters (*c.f.* Figure 1). To bridge the gap, we introduce a learning-based framework, where a parameterized aggregator takes the intra-communication client parameters into consideration, and learns to calibrate the direction of aggregated parameters. Technically, we propose a novel aggregation strategy, named FEDPA, which frames the learning-to-aggregate procedure as a meta-learning setting Ravi & Larochelle (2016); Andrychowicz et al. (2016). In particular, the aggregator is considered as a *meta-learner* that is learning to aggregate the parameters of clients into a proxy model that could generalize well on a proxy dataset. The aggregation process at each communication round refers to one meta-task. The meta-knowledge refers to how to capture the global view under the client/period drift, alleviate the aggregation bias, and calibrate the aggregated parameters towards the optimum.

## 2 RELATED WORK

**Federated learning with non-iid data**  Federated Learning with non-iid Data The performance of federated learning often suffers from the heterogeneous data located over multiple clients. (Zhao et al., 2018) demonstrates that the accuracy of federated learning reduces significantly when models are trained with highly skewed non-iid data, which is explained by weight divergence. (Li et al., 2020) proposes FEDPROX that utilizes a proximal term to deal with heterogeneity. (Li et al., 2021b) provides comprehensive data partitioning strategies to cover the typical non-iid data scenarios. Fed-Nova (Wang et al., 2020) puts insight on the number of epochs in local updates and proposes a normalized averaging scheme to eliminate objective inconsistency. FedBN (Li et al., 2021c) focuses on the feature shift non-iid in FL, and proposes to use local batch normalization to alleviate the feature shift before averaging models.

**Meta Learning** Meta learning is a branch of machine learning in which automated learning algorithms, whose major objective is to use such information to understand how automated learning may become flexible in handling learning problem, thus improving the performance of existing learning algorithms or learning the learning algorithm itself. Meta learning is to solve a problem known as learning to learn (Pramling, 2012), and has shown its effectiveness in reinforcement learning (Xu et al., 2018), few-shot learning (Nichol et al., 2018), image classification (Ravi & Larochelle, 2016). Andrychowicz et al. (Andrychowicz et al., 2016) propose to replace hand-designed update rules with a learned update rule and adopts deep neural networks to train a meta learner by means of an optimizer-optimizee setup, and each component is learnt iteratively using gradient-descent. Also, Ravi (Ravi & Larochelle, 2016) proposes an LSTM meta-learner to learn an optimization procedure as a model for few-shot image classification. Finn et al. propose a Model-agnostic meta-learning (MAML) method (Finn et al., 2017) that does not impose a constraint on the architecture of the learner. Then, derived from MAML, Reptile (Nichol et al., 2018) simplifies the its learning process by conducting first-order gradient updates on the meta-learner.

**Federated Meta Learning** Meta learning plays important roles in federated learning from different aspects, including but not limited to fast adaption, continual learning, personalization, robustness, and computing or communication efficiency. Jiang et al. (Jiang et al., 2019) point out that the setting of MAML, where one optimizes for a fast, gradient-based, few-shot adaptation to a heterogeneous distribution of tasks, has a number of similarities with the objective of personalization for FL, and observe that conventional FEDAVG can be interpreted as a meta learning algorithm. Li (Li et al., 2021a) proposes Meta-HAR that train a shared embedding network can generalize to any individual users, achieving robust learning and personalization. Fallah et al. (Fallah et al., 2020) studies a personalized variant of the federated learning, whose goal is to find an initial shared model that current or new users can easily adapt to their local dataset by performing one or a few steps of gradient descent with respect to their own data. Lin (Lin et al., 2020b) designs a novel federated learning framework for rating prediction (RP) for mobile environments, and employ a meta recommender (MR) module to generate private item embeddings and a RP model based on the collaborative vector. We list other methods that adopt meta learning to federated learning. Recently, (Shamsian et al., 2021) proposes learning a central hypernetworks that acts on client representation vectors for generating personalized models. (Yao et al., 2019) presents FEDMETA that using a proxy dataset for unbiased model aggregation by meta update on server. However, this method updates the global model by directly training on proxy dataset. In our experiments, it has risks at overfitting on proxy dataset.

## 3 METHODOLOGY

The aim of typical federated learning is to learn a shared model over decentralized data. In the federated setting, data cannot be collected in central server and should be locally fixed on various devices, to protect data privacy. FEDAVG is a typical FL method that aggregates local model updates using a weighted averaging strategy, i.e., $w^{global} \leftarrow \sum_{k=1}^{K} \frac{n_k}{n} w^k$. However, FEDAVG suffers from a severe accuracy degradation issue in the non-iid case, i.e. $\mathcal{P}(x, y) \sim \mathcal{P}_k(x, y) \neq \mathcal{P}_j$. In this section, we explore the non-iid problem of typical FEDAVG , and propose a novel framework based on meta learning to deal with it.

### 3.1 TYPICAL FEDERATED LEARNING SETUP

FEDAVG is to learn a single shared model over decentralized data to minimize the global objective $f(w) = \frac{1}{n} \sum_{i=1}^{n} f_i(w)$ in the distributed manner. The objective is the sum of the loss function of all private data $\mathcal{D}_{private}$, that independently generated by a distinct distribution $\mathcal{P}_k(x, y)$ from $K$ clients. The union of decentralized private data forms the training dataset of FL. To minimize the global objective, FEDAVG starts with copying the global model parameters $w_t^k \in \mathbb{R}^d$ by a set of candidate clients, and each candidate then conducts local update that optimizes the local objective by gradient decent method for several epochs:

$$F_k(w_t^k) = \frac{1}{n_k} \sum_{i \in \mathcal{P}_k} f_i(w_t^k), \qquad w_t^k \leftarrow w_t^k - \eta \nabla F_k \left( w_t^k, \mathcal{D}_{private}^k \right), \tag{1}$$

where $F_k(w_t^k)$ is the local objective of the $k$-th client, $n_k$ is the number of local samples, $\eta$ is the local learning rate, and $\nabla F_k\left(w_t^k\right) \in \mathbb{R}^d$ is the gradient vector. After a period of local updates, clients transmit local model parameters $w_t^k$ to the server, who then aggregates these parameters by weighted averaging:

$$w_{t+1}^{global} \leftarrow \sum_{k=1}^{K} \frac{n_k}{n} w_t^k, \tag{2}$$

where $w_{t+1}^{global}$ is the parameters of the global model. Repeat this training process until the global model gets convergence, and the shared global model are collaboratively trained without sharing private data.

However, the expectation $\mathbb{E}_{\mathcal{P}_k}\left[F_k(w)\right] \neq f(w)$ since the data distribution of the $k$-th client may be different with that of any $j$-th client ($\mathcal{P}_k \neq \mathcal{P}_j \neq \mathcal{P}_{overall}$), as well as the overall data distribution in the non-iid setting, leading to client drift (or weight divergence). In spite of some works Li et al. (2020); Karimireddy et al. (2021) that deal with this, they do not considered the problem of "period drift", that is, the data distribution of randomly selected candidate clients of the $t$-th communication round are different with that of the $t + 1$-th communication round, as well as the overall distribution ($\mathcal{P}^t(x, y) \neq \mathcal{P}^{t+1}(x, y) \neq \mathcal{P}_{overall}$). It will also lead to bad convergence.

### 3.2 Parameterized Feedback Aggregator

In this section, we introduce the proposed FEDPA , and explain how it deals with the aggregation bias (client drift as well as period drift), and implement it within a meta learning framework.

Inspired by control theory, we naturally consider federated learning as a dynamic system Haddad & Chellaboina (2011), where we regard model parameters $w_t$ as system states. Actually, the training process of FEDAVG is an autonomous system that without control, and its differential equations (1)(2) can be written as follows [2]:

$$\begin{aligned} w_{t+1} &= g(w_t) \\ &= \frac{1}{K} \sum_{k=1}^{K} w_t^k = \frac{1}{K} \sum_{k=1}^{K} (w_t - \Delta w_t^k). \end{aligned} \tag{3}$$

where $g(w_t)$ includes the model parameters $w_t$ and the local update $\Delta w_t^k$ that minimizes $F_k(w_t^k)$ for several epochs, as well as parameters averaging, who determines the trajectory of model parameters $w_t$. However, due to the non-iid data, the objective $F_k(w_t^k)$ of selected candidates could be an arbitrarily poor approximation to the global objective $f(w_t)$, resulting in aggregation bias (client drift and period drift). Our idea is to control the local update approaching to the optimum by adding a control variable $u_t^k$ for each client, intervening the trajectory of FL. We formulate the controlled system $g_c(w_t)$ as follows:

$$\begin{aligned} w_{t+1} &= g_c(w_t) \\ &= \frac{1}{K} \sum_{k=1}^{K} (w_t - \Delta w_t^k(1 - u_t^k)) = \frac{1}{K} \sum_{k=1}^{K} (w_t - \Delta w_t^k(1 - h(w_t, \Delta w_t^k, \phi))) \end{aligned} \tag{4}$$

where we define $u = h(\cdot, \phi)$ as the controller of $\Delta w_t^k$, parameterized by $\phi$, who takes the model parameters $w_t$ and $\Delta w_t^k$ as inputs. Now, we package the averaging operator and the controller as the aggregator, and define the aggregation function as $aggr(w_t, \Delta \mathcal{W}_t, \phi) = \frac{1}{K} \sum_{k=1}^{K} (w_t - \Delta w_t^k(1 - h(w_t, \Delta w_t^k, \phi)))$, where $\Delta \mathcal{W}_t = \{\Delta w_t^k\}$ is the set of local updates of $t$-th candidates. Finally, the differential equation is formulated as:

$$w_{t+1} = aggr(w_t, \Delta \mathcal{W}_t, \phi). \tag{5}$$

The question now is *how can we get an effective aggregator to debias model aggregation?* We implement it within a meta learning framework. Inspired by Andrychowicz et al. (2016); Ravi &

---

[2]In fact, the number of samples of clients $n_k$ is usually unknown to server, thus we set $\frac{n_k}{n}$ as $\frac{1}{K}$.

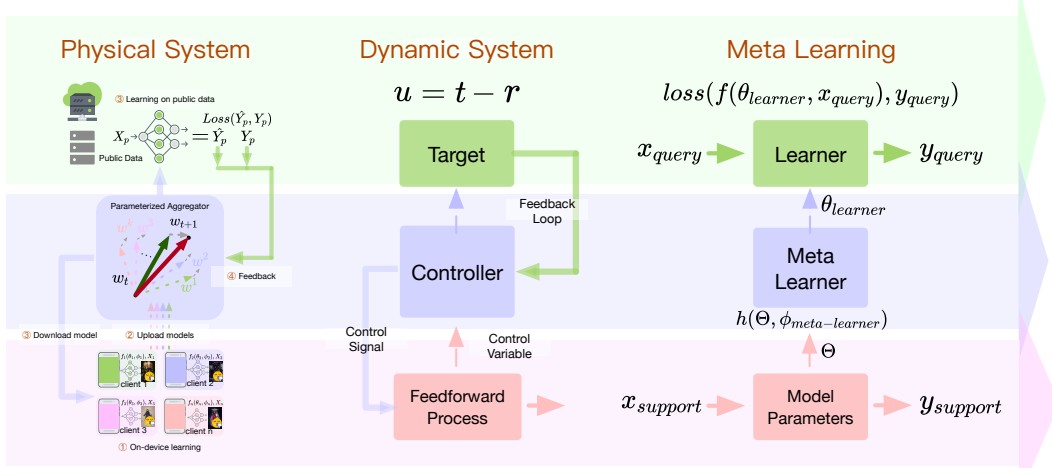

Figure 2: **Relations of FEDPA , dynamic system and meta learning.** The left figure shows the pipeline of FEDPA , including local updates, model aggregation and the aggregator of training. We explain the process of FL as a dynamic process of $w_t$, and the trajectory of $w_t$ should be controlled towards global optimum due to the aggregation bias (client drift and period drift). Thus, the performance of aggregated model on the proxy dataset is the target in the control loop, and the parameterized aggregator is the controller, shown in the middle figure. To achieve the training of aggregator (controller), we frame it within a meta learning setting, where the aggregator is a meta learner that helps the aggregated model (learner) to achieve good performance on the proxy dataset.

Larochelle (2016), clients in FL can be regarded as *learners* whose their privacy data are considered as support set. At the meantime, the aggregator is a *meta-learner*, and we utilize a set of proxy data as query set. The Relations of FEDPA , dynamic system and meta learning are shown in Figure 3.2.

The training of aggregator is implemented by testing the performance of an aggregated model on the proxy dataset. At each communication round, the server receives and aggregates clients' parameters into a proxy model, and then evaluates the performance of proxy model on the proxy dataset. We assume that the better performance proxy model has, the better aggregator becomes. Thus, we can optimize the aggregator by the objective as follows:

$$\min_{\phi} f(w_{proxy}, \mathcal{D}_{proxy}),$$

$$\text{where} \quad w_{proxy} = aggr(w_t, \Delta \mathcal{W}_t.\phi). \tag{6}$$

For the model parameters $w$ with $m$ layers $w = \{w_{1 \cdots m}\}$, the parameters of aggregator $\phi$ consist of two sets of input dense layers and one set of output layers $\phi = \{\phi_{1 \cdots m}^{w\_in}, \phi_{1 \cdots m}^{\Delta w\_in}, \phi_{1 \cdots m}^{out}\}$. Considering the dimension explosion problem ($\phi \in \mathbb{R}^{d \times d}$ if $w \in \mathbb{R}^d$), we design the network with the bottleneck architecture, mapping the parameters to a low-dimension space, and finally restore the output to the original dimension. For example, we first respectively input $\{w_{1 \cdots m}\}$ and $\{\Delta w_{1 \cdots m}\}$ (with dimensions $d_{1 \cdots m}$) into the dense layers $\{\phi_{1 \cdots m}^{w\_in}\}$ and $\{\phi_{1 \cdots m}^{\Delta w\_in}\}$, and output two low-dimension hidden states $h_{1 \cdots m}^{w}, h_{1 \cdots m}^{\Delta w}$ (with dimensions $p_{1 \cdots m} = log_2(d_{1 \cdots m}) + 1$). We then concatenate $\{h_{1 \cdots m}^{w}, h_{1 \cdots m}^{\Delta w}\}$ and finally input it into output dense layers $\{\phi_{1 \cdots m}^{out}\}$ to restore the original dimension, becoming the control variable $u_t^k$:

$$u_t^k = h(\phi, w_t^k)$$
$$= dense(concat(dense(w_{1 \cdots m}, \phi_{1 \cdots m}^{w\_in}), dense(\Delta w_{1 \cdots m}, \phi_{1 \cdots m}^{\Delta w\_in}))\phi_{1 \cdots m}^{out}), \tag{7}$$

and note that all these operators are layer-wisely for each local model $w_t^k$.

The whole process shows in algorithm 1. At the training stage, just like the usual FL, 1) the server randomly samples a set of candidates $\mathcal{K}_t$ and starts clients local update and upload their updated

---

**Algorithm 1** FEDPA : require the global model $w_t$, the proxy dataset $\mathcal{D}_{proxy}$ on server, the clients indexed by $k$ and their local model $w_t^k$ and private dataset $\mathcal{D}_{private}^k$, the local learning rate $\eta_l$, the number of local epochs $E_l$ and the number of epochs for training aggregator $E_g$, and the total number of rounds $T$.

---

**Server executes:**
  1: initialize the global model $w_0$
  2: **for** each round $t = 0, 1, 2, \ldots, T$ **do**
  3:     randomly sample a set of candidate clients $\mathcal{K}$
  4:     *at the meantime:*
  5:     a) $\Delta \mathcal{W}_t \leftarrow$ ClientsUpdate$(w_t, \mathcal{K})$,
  6:     b) update the aggregator by equation 6 for $E_g$ epochs}
  7:     $w_{t+1} = aggr(w_t, \Delta \mathcal{W}_t, \phi)$
  8: **end for**

**ClientsUpdate:**
  1: **for** each client $k \in \mathcal{K}$ **in parallel do**
  2:     download: $w_t^k \leftarrow w_t$
  3:     **for** each epoch $e = 1, 2, \ldots, E_l$ **do**
  4:         Conduct local update: $w_t^k \leftarrow w_t^k - \eta_l \nabla F_k \left( w_t^k, \mathcal{D}_{private}^k \right)$
  5:     **end for**
  6:     upload: $\Delta w_t^k \leftarrow w_t - w_t^k$
  7: **end for**
  8: $\Delta \mathcal{W}_t = \{ \Delta w_t^k \}$

---

model parameters $\Delta \mathcal{W}_t$ back to server; 2) the server aggregates all these parameters into a new global model by equation 5, and start next generation of 1); 2) at the meantime (during clients local update), the server trains the aggregator by equation 6; 3) repeat these steps until FL stops. By learning the $\phi$, the aggregator can capture the ability of calibrating and controlling the aggregated parameters in a global view to a better direction towards optimum, that solves aggregation bias (client drift and period drift).

## 4 EXPERIMENTS

### 4.1 SETUP

**Datasets and models.** We evaluate FEDPA with different state-of-the-art FL methods on both CV and recommendation dataset. For CV dataset, we use FEMNIST[3] Caldas et al. (2018), consisting of 671,585 training examples and 77,483 test samples of 62 different classes including 10 digits, 26 lowercase and 26 uppercase images with 28x28 pixels, handwritten by 3400 users. It is an image classification task, and we use the lightweight model LeNet5 LeCun et al. (1998). For recommendation dataset, we use MovieLens 1M [4]Harper & Konstan (2015) (including 1,000,209 ratings by unidentifiable 6,040 users on 3,706 movies. It is a click-through rate (CTR) task, and we use the popular DIN Zhou et al. (2018) model. For performance evaluation, we follow a widely used leave-one-out protocol Muhammad et al. (2020). For each user, we hold out their latest interaction as testset and use the remaining data as trainset, and binarize the user feedback where all ratings are converted to 1, and negative instances are sampled 4:1 for training and 99:1 for test times the amount of positive ones.

**Federated learning settings.** Note that the datasets we use (both FEMNIST and MovieLens 1M) have "natural" non-iid distribution since we can split the dataset by userid, i.e. image data are handwritten by different users and movies are rated by different users. Beside, we use the Dirichlet distribution Hsu et al. (2019) to simulate the label distribution skew setting for FEMNIST, where the hyperparameter $\alpha$ controls the degree of non-iidness. The smaller $\alpha$, the more degree of non-iid distribution. For FL training, we set totally $T = 100$ communication rounds, and sample 10% of all

---

[3]https://github.com/TalwalkarLab/leaf/tree/master/data/femnist (BSD-2-Clause license)
[4]https://grouplens.org/datasets/movielens/ (license)

clients per round and each client trains $E_l = 5$ epochs at local update, using Adam optimizer Kingma & Ba (2014) with learning rate $\eta_l = 0.01$. In our proposed FEDPA , we use the proxy dataset that are randomly sampled $1\%$ from training data for all tasks. As for the training of the aggregator, we set $E_g = 5$ for MovieLens and $E_g = 30$ for FEMNIST, with learning rate $\eta_g = 0.001$.

**Baselines.** FEDPA is a server-side method that improves the model aggregation, thus we compare FEDPA with 1) the vanilla FL method FEDAVG McMahan et al. (2017), 2) a client-side FL method FEDPROX Li et al. (2020), 3) a server-side FL method without using proxy data FEDAVGM Hsu et al. (2019), 4) a server-side FL method without using proxy data FEDOPT Reddi et al. (2020), 5) a server-side FL method with proxy data FEDDF Lin et al. (2020a), 6) a server-side federated meta learning method with proxy data FEDMETA Yao et al. (2019). Note that FEDPA is to solve the slow convergence of training, thus we omit other excellent FL method involving meta learning designed for model initialization or fast adaption Chen et al. (2018), for personalization Shamsian et al. (2021); Fallah et al. (2020).

**Evaluation Metrics.** For image classification task, model performance in our experiments are measured by the widely used top-1 accuracy. For CTR task, model performance are measured by some popular metrics: area under curve (AUC), Hit Ratio (HR) and Normalized Discounted Cumulative Gain (NDCG):

$$\text{AUC} = \frac{\sum_{x_0 \in D_T} \sum_{x_1 \in D_F} \mathbf{1}\left[f\left(x_1\right) < f\left(x_0\right)\right]}{|D_T||D_F|},$$

$$\text{HitRate@K} = \frac{1}{|\mathcal{U}|} \sum_{u \in \mathcal{U}} \mathbf{1}\left(R_{u,g_u} \leq K\right),$$

$$\text{NDCG@K} = \sum_{u \in \mathcal{U}} \frac{1}{|\mathcal{U}|} \frac{2^{\mathbf{1}\left(R_{u,g_u} \leq K\right)} - 1}{\log_2\left(\mathbf{1}\left(R_{u,g_u} \leq K\right) + 1\right)},$$

where $\mathcal{U}$ is the user set, $\mathbf{1}$ is the indicator function, $R_{u,g_u}$ is the rank generated by the model for the ground truth item $g_u$ and $f$ is the model to be evaluated and $D_T, D_F$ is the positive and negative sample sets in testing data.

**Implementation** The experiments are implemented with PyTorch. We simulate the FL environment including clients and run all experiments on a deep learning a server with NVIDIA Tesla V100.

## 4.2 ANALYSIS

**Visualize the impact of period drift.** We conduct this experiments on the FEMNIST dataset with three degrees of non-iidness, that we set the dirichlet hyperparameter $\alpha = 1/0.1/0.01$. The top three figures visualize the degree of non-iidness with different $\alpha$. We select 10 candidates from 3400 clients, and show the label distribution of the 10-digits labels among 62 classes for the 20 earliest communication rounds. From left to right, we show that the size of points becomes more diverse both within the same communication round and between different communication rounds, that exhibits client drift and period drift, respectively. For the bottom three figures, we show the impact of period drift by five curves for each setting. The FL training becomes more difficult as increasing the degree of non-iidness. Especially note the setting of "iid_each_round". To show the impact of period drift, we manually eliminate the impact of client drift by shuffling the data of selected candidates at each communication round, to force the data become iid. The figures show that not only the client drift, period drift will lead to slow convergence and oscillating, and proposed FEDPA can achieve fast and steady convergence especially with smaller $\alpha$.

**The performance on MovieLens.** We conduct this experiment on the MovieLens 1M dataset, which is naturally non-iid, and evaluate it by auc, hit@5, hit@10, ndcg@5, ndcg@10. As Table 4.2 shown, our proposed FEDPA outperforms alternatives on most metrics. Due to the well trained aggregator, our FEDPA has the fastest convergence than other sota methods. Methods that without proxy dataset, have low convergence because samples of CTR task is kind of strong non-iid since each user have distinct user profiles like userid, age, sex and so on. Besides, each user rates limited movies, which leads to a few amount of embedding table in the model being updated. For methods

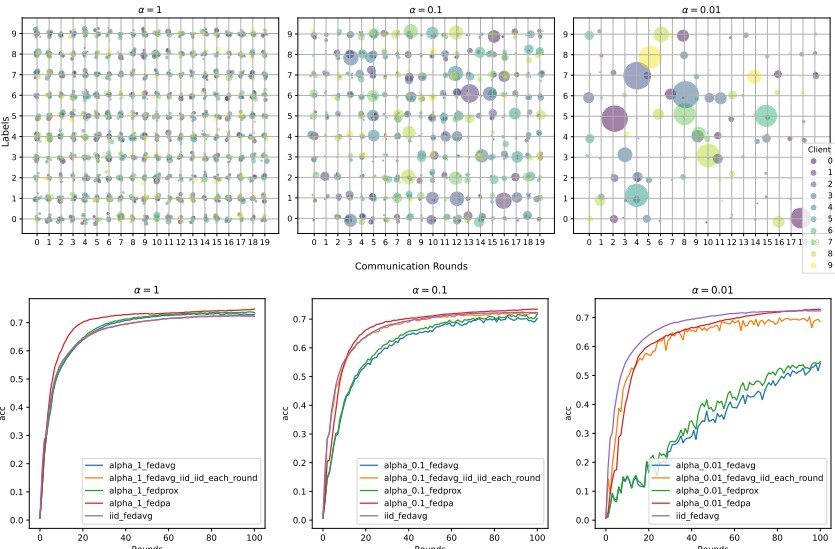

Figure 3: **The impact of Period drift.** This experiments is on the FEMNIST dataset with three degrees of non-iidness. For the top figures, we select 10 clients and choose the 10-digits labels. In each scatter plot, the x axis is twenty communication rounds and the y axis is ten labels. A specified point is a specified client has samples that of a specified label at a specified communication round, whose size means the number of samples and color means the client it belongs to. The size of points becomes more diverse within the same communication round or between different communication rounds, that exhibits client drift and period drift, respectively. For the bottom figures, we show the impact of period drift by five curves for each setting. As the figures show, not only the client drift, period drift will lead to slow convergence, especially with smaller $\alpha$.

that use proxy dataset, comparing with FEDPA , FEDDF and FEDMETA have limited performance but have different results and for different reasons. FEDDF has no advantage for the CTR task since the model is of a logistic regression that only has one output, thus FEDDF can hardly benefit from ensemble distillation on proxy dataset. FEDPA can achieve higher performance than other methods since it can get into a smaller optimum since FEDPA can deal with the period drift, which has different objectives at different communication round even around global optimum.

**The performance on FEMNIST.**   We conduct this experiment on the FEMNIST dataset, and we use four settings to evaluate the performance. As Table 4.2 shown, our proposed FEDPA outperforms alternatives. As increasing the degree of non-iidness, proposed method shows little performance degradation, comparing to other FL method. FEDPA benefits from the well-trained aggregator that can calibrate model parameters in a global view, which resulting in good performance at extreme non-iid setting. Compare with other FL methods, FEDPA can achieves fast and steady convergence, and a better optimum.

**The advatages of learning-based FEDPA**   Because of the "no free lunch" (NFL) theorem, it is hard to have a perfect method that appropriate to all datasets and scenarios. However, proposed FEDPA provides a framework that can target to a specific problem. The ability of the aggregator is from the proxy datasets, which is specific and adaptive to a certain task. The aggregator can learn to aggregate for different model, datasets and even the non-iidness.

**The difference of using proxy dataset.**   We compare FEDPA with the baselines that using proxy dataset, FEDDF and FEDMETA . The common ground of these three methods are they all use the proxy dataset to achieve fast convergence and better performance. However, the way of using proxy dataset and the reason of how it helps FL are very different. FEDDF uses unlabeled proxy data by leveraging ensemble distillation, taking advantage of ensembling the diverse model parameters of clients as well as distilling the logits to achieve consensus. FEDMETA uses proxy dataset by meta

updating that training global model at each communication round. However, the problem of these two method is that they have the risks of overfitting on the proxy dataset, since they directly update the model parameters, even carefully tune the hyperparameter of training epochs and regularization. Instead, FEDPA does not directly update model parameters but learns a aggregative bias to control the training process of FL. Thus, FEDPA is more safe than the methods that directly updating model parameters.

Table 1: **The performance on MovieLens 1M**

|  | **Auc** | **Hit@5** | **Hit@10** | **Ndcg@5** | **Ndcg@10** |
| --- | --- | --- | --- | --- | --- |
| FEDAVG | 0.7482 | 0.2916 | 0.4290 | 0.1901 | 0.2346 |
| FEDAVGM | 0.7482 | 0.2916 | 0.4290 | 0.1901 | 0.2346 |
| FEDPROX | 0.7459 | 0.2924 | 0.4298 | 0.1914 | 0.2358 |
| FEDOPT | 0.7250 | 0.2967 | 0.4419 | 0.1904 | 0.2374 |
| FEDDF | 0.7053 | 0.2553 | 0.3623 | 0.1701 | 0.2046 |
| FEDMETA | 0.7651 | 0.2930 | **0.4429** | 0.1919 | 0.2404 |
| FEDPA | **0.7878** | **0.3058** | 0.4382 | **0.2002** | **0.2431** |

Table 2: **The performance on FEMNIST**

|  | **Natural** | $\alpha = 1$ | $\alpha = 0.1$ | $\alpha = 0.01$ |
| --- | --- | --- | --- | --- |
| FEDAVG | 0.6909 | 0.7299 | 0.7029 | 0.5427 |
| FEDAVGM | 0.6909 | 0.7299 | 0.7029 | 0.5427 |
| FEDPROX | 0.6990 | 0.7360 | 0.7177 | 0.5478 |
| FEDOPT | 0.6830 | 0.7295 | 0.7156 | 0.5157 |
| FEDDF | 0.6921 | 0.7311 | 0.6955 | 0.5271 |
| FEDMETA | 0.6967 | 0.7316 | 0.7194 | 0.5617 |
| FEDPA | **0.7444** | **0.7431** | **0.7261** | **0.7224** |

## 5 PRIVACY CONCERNS

As for the proxy dataset, we understand this is denounced in some situations, since it may violate the constraint of privacy because there are some scenarios that have no proxy dataset like healthcare. However, in some scenarios that allows proxy dataset like natural images recognition, item interactions with personal information wiped, proxy dataset can help FL to a large extent Li & Wang (2019); Lin et al. (2020a); Zhang et al. (2021); Yao et al. (2019) etc. Instead, we encourage the use of proxy dataset in FL (if exists), since many companies like Google, Facebook remains previous data at the turning point of legislation for privacy, and how to use the proxy dataset is also an interesting problem.

## 6 CONCLUSION

In this work, we provide an another factor to degrade the convergence of cross-device FL, namely, the period drift. To solve the period drift as well as the client drift, we propose a novel aggregation strategy, FEDPA , as an alternative of averaging. We analyze the problem in a view of dynamic system and frame the training procedure of FEDPA within a meta-learning setting. Experiments show that the proposed method outperforms other state-of-the-art methods, indicating that the trained aggregator can well capture the ability of aggregating drifted clients' model parameters. It also has a global view to alleviate the period drift, by adding a parameter-wise bias for each client to calibrate and control the aggregated parameters to a better direction towards the optimum.

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
