# OpenReview forum: "Learning to aggregate: A parameterized aggregator to debias aggregation for cross-device federated learning"
_ICLR.cc/2023/Conference — Submitted to ICLR 2023_

### Official Review · Reviewer_5zVx · 2022-10-24

**Confidence:** 3
**Clarity, Quality, Novelty And Reproducibility:** 0.Could you explain how the drift iss…
**Correctness:** 3
**Technical Novelty And Significance:** 3
**Empirical Novelty And Significance:** 2
**Recommendation:** 6

**Strength And Weaknesses:**

Strength
1. The 2 levels of drift they tackle is indeed important for FL.
2. They can achieve better training performance on the MovieLens dataset and FEMNIST dataset.
Weaknesses
1. Some small typos, grammar issues, inconsistency issues as stated below.
2. The key idea should be agnostic to the dataset and types of tasks. It would be good to show the performance on other LEAF datasets.
3. It would be good if the overhead on the server can be quantified. If the method can not be applied at scale (equation 7 seems to be iterative but more clarification would be good), it is not a perfect match for FL.


**Summary Of The Paper:**

1. The paper tackles the problem of client drift (locally) in federated learning (FL) due to heterogeneous client data distributions. Besides, they target the period drift (globally), which is the inter-communication heterogeneity of data distributions.
2. They propose a learning based parameterized aggregator called FEDPA, debiasing model aggregation under client drift and period drift in a unified framework. Their key approach is to learn an adaptive calibration parameter to approximate the global objective.
3. The input of the framework includes the intra-communication client parameters.


**Summary Of The Review:**

The introduction and related work part is in good shape and I enjoyed reading it. But the quality for writing could be improved. (See the above session for more details)

---

### Official Review · Reviewer_o7iB · 2022-10-25

**Confidence:** 4
**Correctness:** 3
**Technical Novelty And Significance:** 3
**Empirical Novelty And Significance:** 2
**Recommendation:** 3

**Clarity, Quality, Novelty And Reproducibility:**

AFAIK, the idea is novel.

Need improvement and clarification: the intra/inter-communication arguments look inaccurate to me. The global model and control variates are shared “inter” rounds, for example, in SCAFFOLD [Karimireddy et al. 2021]. Some previous work also assume all clients can participate in training, and I would strongly encourage the authors to clarify the source of “intra-round heterogeneity”

Could you clarify “since many companies like Google, Facebook remains previous data at the turning point of legislation for privacy” for motivating the proxy dataset?

I may have missed it, is the code open sourced? The authors mention they implement the algorithms in PyTorch. Using a FL framework, or based on previous released code can significantly help reproducibility.

Minor issue:
The citation format does not seem to be consistent. I would suggest the authors carefully consider the usage of `\citep` and `\citet`.
I cannot understand why Kairouz et al 2021 is cited for Figure 1.
Some grammatical errors might need to be corrected.


**Strength And Weaknesses:**

The general idea of “meta learning” and “learning to aggregate” makes sense.

However, as the authors commented, though interesting, having a proxy dataset is a strong assumption in problem setting.

In addition, the server seems to have access to the model updates of each individual client, which makes it hard to be consistent with FL privacy principles, and other privacy techniques like SecAgg and differential privacy [Federated Learning and Privacy https://queue.acm.org/detail.cfm?id=3501293]

My major concern is that the proposed method seems to be ad-hoc. It is hard for me to connect the motivation of “period shift” to the proposed FedPA method. Instead of learning neural networks for aggregation, I am wondering if there are easier approaches to use the proxy data. For example, we can simply “learn” a scalar weight to do weighted aggregating/averaging of client model updates. I would strongly suggest some more ablation studies of the proposed FedPA method.

A second major concern is the experiment performance. The accuracy on FEMNIST seems to be lower than expected. For example, [Adaptive Federated Optimization https://arxiv.org/abs/2003.00295] reports >80% for natural user non-IID.

I would also appreciate some more comments on hyperparameter tuning. For example, how are 100 communication rounds, 5 epochs, learning rate \eta_l=0.01 chosen? How are training epochs (5 for MovieLens, 30 for FEMNIST) and learning reate \eta_g chosen?



**Summary Of The Paper:**

This paper proposed a method named FedPA to learn aggregators in federated learning. When aggregating model updates from clients, instead of uniform or weighted by number of examples as in the popular FedAvg, FedPA will feed both the global model and the client model updates to a neural network before “aggregating”/averaging. The aggregator is trained on the server with a proxy dataset. Experiments on EMNIST and MovieLens show the advantage of FedPA.


**Summary Of The Review:**

The idea is interesting, but the draft itself needs improvement. Ablation study and experimental performance are my main concerns.

---

### Official Review · Reviewer_L63q · 2022-10-31

**Confidence:** 4
**Correctness:** 3
**Technical Novelty And Significance:** 2
**Empirical Novelty And Significance:** 3
**Recommendation:** 5

**Clarity, Quality, Novelty And Reproducibility:**

This paper is well-motivated and easy to follow. The technical novelty is ok but not much. It is also ambiguous whether the proposed method is the best solution for this problem (please see weakness).

**Strength And Weaknesses:**

Strength:
1. Drifts in FL arise in time and space, while most existing works only address the heterogeneity of client data distributions. This paper has discovered this practically important problem and proposed the notion of period drift that can facilitate further research.
2. The authors have conducted comprehensive experiments in different settings. Results have shown that FedPA has a superior advantage over baselines in different categories.

Weakness:
1. This paper lacks an in-depth discussion on why meta-learning frameworks are particularly suited for the period drift problem. It seems like both client and period drift influence the model performance by introducing extra bias in model aggregation. In that case, why not use a regularization-based approach incorporated with a temporal dimension? Moreover, it seems like this paper [1] have studied a similar problem, the authors could consider comparing FedPA with their work as an additional baseline.
2. The dynamic system analogy seems useless in section 3. The authors are not using significant knowledge from this area. I would recommend adding more discussions or simply removing this part to avoid confusion.
3. From my understanding, FedPA accounts for additional bias via controlling $\Delta w_t^k$ through $u_t^k$, then why do we need two separate neural networks for both $w$ and $\Delta w$? The authors need to be more specific on the choice of NN architectures.

Minor:
1. Please add a discussion on FedDL in section 2.
2. Please move the definition of $n_k$ and $n$ to the beginning of section 3.

[1] Jothimurugesan et al., Federated Learning under Distributed Concept Drift, 2022

**Summary Of The Paper:**

This paper proposed a method called FedPA that deals with client and period drift problems. The period drift problem is caused by the asynchronized updates of each client, leading to extra bias in model aggregation. The authors proposed a learning-based aggregation strategy, that parameterizes the aggregation function using neural network models. The models are trained under a meta-learning framework, which treats the global model as a meta-learner and each client as a specific task. Experimental results have shown that FedPA can account for the additional bias induced by both client and period drift and therefore demonstrate superior performance over other FL baselines in various tasks.

**Summary Of The Review:**

This work can benefit from adding more in-depth discussion on the unique advantages of the proposed method and further polishing the writing. I am leaning toward rejecting this paper at this time.

---

### Official Review · Reviewer_7sGj · 2022-11-02

**Confidence:** 4
**Correctness:** 2
**Technical Novelty And Significance:** 1
**Empirical Novelty And Significance:** 2
**Recommendation:** 3

**Clarity, Quality, Novelty And Reproducibility:**

Clarity: The paper is not very well written and has some grammatical mistakes.
Quality: The paper quality needs to be improved. The axes font in the figures is too small to read and overall the writing needs to be updated.
Novelty: The paper has limited novelty.
Reproducibility: No code was given.

**Strength And Weaknesses:**

Strengths
1. The paper identifies a possible source of client drift
2. The paper proposes a novel aggregation scheme.

Weaknesses
1. The paper does not do enough to discriminate between regular client drift and the so called period drift either theoretically or through experiments.
2. The aggregation strategy uses a proxy dataset which limits use cases. Also, it is very similar to other knowledge distillation-based techniques like FedET[1] and DS-FL[2]. A comparison of performance with these methods should be shown to justify its usefulness.
3. There is no ablation study showing the effect of the data distribution in the proxy data on model performance.
4. The experimental settings are not strong. The datasets and models are too simple. I suggest including results on CIFAR-100 and Stack Overflow datasets.


[1] Cho, Y. J., Manoel, A., Joshi, G., Sim, R., & Dimitriadis, D. (2022). Heterogeneous Ensemble Knowledge Transfer for Training Large Models in Federated Learning. arXiv preprint arXiv:2204.12703.

[2] Itahara, S., Nishio, T., Koda, Y., Morikura, M., & Yamamoto, K. (2020). Distillation-based semi-supervised federated learning for communication-efficient collaborative training with non-iid private data. arXiv preprint arXiv:2008.06180.


**Summary Of The Paper:**

The paper presents a learnable aggregation scheme in the context of federated learning. The paper achieves this using meta-learning to generalize the parameters of the aggregator with a proxy dataset. The paper identifies 'period drift' in the current federated learning setup and presents the meta-learning-based aggregator as a way to overcome this issue. The paper follows up with experimental results showing increased accuracy for different methods and heterogeneity rates across two datasets.

**Summary Of The Review:**

The paper proposes a meta-learning-based aggregation scheme. However, it does not show enough theoretical or experimental justification to highlight the effectiveness of the algorithm. Additionally, the paper lacks enough ablation studies on the different aspects of the algorithm like the data distribution of proxy data, the influence of the size of the aggregator model, etc.  Furthermore, the paper's concept of 'period drift' is not well defined despite being a key motivation of the algorithm.

---

### Decision · Program_Chairs · 2023-01-20

**Decision:**

Reject

**Justification For Why Not Higher Score:**

All reviewers have severe concerns, and authors have not answered

**Justification For Why Not Lower Score:**

N/A

**Metareview: Summary, Strengths And Weaknesses:**

The paper studies a combination of client drift and period drift (over time, coming from asynchronous updates) in the setting of federated learning. A learning-based aggregation strategy is proposed, and learned using meta-learning on a proxy dataset.

Unfortunately many concerns remained from the reviews both on fundamentals and experiments, and no author feedback was given.

We hope the detailed feedback helps to strengthen the paper for a future occasion.